# The Variations’ in Genes Encoding TIM-3 and Its Ligand, Galectin-9, Influence on ccRCC Risk and Prognosis

**DOI:** 10.3390/ijms24032042

**Published:** 2023-01-20

**Authors:** Anna Andrzejczak, Krzysztof Tupikowski, Anna Tomkiewicz, Bartosz Małkiewicz, Kuba Ptaszkowski, Aleksandra Domin, Tomasz Szydełko, Lidia Karabon

**Affiliations:** 1Laboratory of Genetics and Epigenetics of Human Diseases, Department of Experimental Therapy, Hirszfeld Institute of Immunology and Experimental Therapy, Polish Academy of Sciences, 53-114 Wrocław, Poland; 2Subdivision of Urology, Lower Silesian Center for Oncology, Pulmonology and Hematology, 53-413 Wrocław, Poland; 3University Center of Excellence in Urology, Department of Minimally Invasive and Robotic Urology, Wroclaw Medical University, 50-556 Wrocław, Poland; 4Department of Clinical Biomechanics and Physiotherapy in Motor System Disorders, Wrocław Medical University, 50-556 Wrocław, Poland

**Keywords:** TIM-3, HAVCR2, galectin-9 (*LGALS9*), clear cell renal cell carcinoma (ccRCC), single-nucleotide gene polymorphism (SNP), immunological checkpoint, disease risk, overall survival

## Abstract

Renal cell cancer is the most common type of kidney cancer in adults, and clear cell renal cell carcinoma (ccRCC) is the most diagnosed type. T cell immunoglobulin and mucin-domain-containing-3 (TIM-3) belongs to immunological checkpoints that are key regulators of the immune response. One of the known TIM-3 ligands is galectin-9 (*LGALS9*). A limited number of studies have shown an association between *TIM-3* polymorphisms and cancer risk in the Asian population; however, there is no study on the role of *LGALS9* polymorphisms in cancer. The present study aimed to analyze the influence of *TIM-3* and *LGALS9* polymorphisms on susceptibility to ccRCC and patient overall survival (OS), with over ten years of observations. Using TaqMan probes, ARMS–PCR, and RFPL-PCR, we genotyped two *TIM-3* single-nucleotide polymorphisms (SNPs): rs1036199 and rs10057302, and four *LGALS9* SNPs: rs361497, rs3751093, rs4239242, and rs4794976. We found that the presence of the rs10057302 A allele (AC + AA genotypes) as well as the rs4794976 T allele (GT + TT genotypes) decreased susceptibility to ccRCC by two-fold compared to corresponding homozygotes. A subgroup analysis showed the association of some SNPs with clinical features. Moreover, *TIM-3* rs1036199 significantly influenced OS. Our results indicate that variations within *TIM-3* and *LGALS9* genes are associated with ccRCC risk and OS.

## 1. Introduction

Renal cell carcinoma (RCC) is the most frequently diagnosed type of renal cancer in adults between 50 and 70 years old, accounting for more than 90% of renal cancer cases. According to WHO data, approximately 431,288 new renal cancer cases were diagnosed in 2020 [1]. The incidence of RCC is two times higher in men than in women. RCC has one of the highest mortality rates of all genitourinary cancers (179,368 deaths in 2021) [1,2]. RCC is divided histologically into three major subtypes: clear cell RCC (ccRCC), papillary RCC, and chromophobe RCC, with ccRCC accounting for 70–80% of all RCC cases [3].

ccRCC is named after its distinctive microscopic imaging, where tumor cells look similar to clear soap bubbles. ccRCC originates from the epithelial cells of the proximal tubule (renal cortex), and in most cases presents a rapidly expansive growth pattern, classifying ccRCC as an aggressive tumor. The development of ccRCC in most cases is sporadic (95%), but in some cases ccRCC is associated with inherited syndromes, including von Hippel–Lindau disease and tuberous sclerosis [3]. One of the reasons for the high mortality of renal cancer is its asymptomatic nature in the early stages of the disease, resulting in patients with metastatic tumors at the time of their diagnosis [4]. Due to a lack of reliable early diagnostic markers of ccRCC, there is a need to identify new diagnostic and prognostic markers for ccRCC.

Immunological checkpoints (ICs) are crucial molecules that maintain immune tolerance and prevent autoimmunity by adjusting the duration and severity of immune responses [5]. Cancer cells learn how to take advantage of these properties of ICs for their own benefit by overexpressing ICs, helping the tumor hide from immune system surveillance [6,7,8]. To date, several IC receptors and their ligands have been identified, including T cell immunoglobulin and mucin-domain-containing molecule-3 (TIM-3). TIM-3, also known as hepatitis A virus cellular receptor 2 (HAVCR2), is a type I transmembrane protein. Various studies have shown that TIM-3 is overexpressed in different types of cancer, such as urothelial carcinoma, prostate cancer, and lung cancer [9,10,11]. Interestingly, TIM-3 is overexpressed on both immune cells and cancer cells. High TIM-3 expression promotes the tumorigenesis, proliferation, and invasion of tumor cells by suppressing immune cells’ functions [12]. TIM-3 binds to several ligands, including galectin-9, CEACAM1, phosphatidylserine (PtdSer), and HMGB1. The first TIM-3 ligand that was discovered was galectin-9 (coded by the *LGALS9* gene), which is widely expressed by various organ systems and tissues. The binding of TIM-3 to galectin-9 initiates inhibitory pathways leading to the suppression of Th1 and Th17 functions that induce immune tolerance [13,14]. Altered galectin-9 expression has been reported in different types of cancers and is negatively correlated with overall survival (OS) in patients [15,16,17,18], making galectin-9 an interesting biomarker and potential target for immunotherapy.

Despite the huge success of immunotherapy, only 20–40% of cancer patients respond to it [19]. Therefore, searching for reliable predictors or markers of cancer development has become an important topic in recent years. Genetic variations are considered to be potential cancer prediction markers. Among them, single-nucleotide polymorphisms (SNPs) have been proven to have an impact on human health and predisposition to certain diseases, including cancer [20,21,22,23]. Polymorphisms of the gene encoding *TIM-3* have previously been reported to be associated with cancer susceptibility and patient OS. Moreover, the presence of specific *TIM-3* SNPs was documented to correlate with TIM-3 expression, modifying cancer risk [24,25,26,27,28]; however, most published studies were conducted on Chinese populations, with a lack of data on Caucasian populations. This being the case, we aim to study the association of *TIM-3* polymorphisms with ccRCC risk in the Caucasian population.

For our study, we selected two *TIM-3* polymorphisms: rs1036199 and rs10057302. rs1036199 has previously been described as being associated with cancer risk and outcomes, but mostly in the Asian population, with only one study concerning RCC. Moreover, we investigated additional polymorphisms within the *LGALS9* gene: rs3751093, rs4239242, and rs4794976, which were previously described in the context of autoimmune diseases [29,30]. All of the studied polymorphisms, as well as their gene localizations, are presented in Figure 1. To the best of our best knowledge to date, *LGALS9* polymorphisms have not been studied in the context of neoplastic diseases, so our study is the first to undertake this problem in ccRCC.

In summary, the aim of the present study is to analyze the influence of selected single-nucleotide polymorphisms in the *TIM-3* and *LGALS9* genes on ccRCC susceptibility and disease progression in the Polish population.

## 2. Results

### 2.1. Association between TIM-3 and LGALS9 SNPs and Susceptibility to ccRCC

Each polymorphism in the *TIM-3* and *LGALS9* genes were in Hardy–Weinberg equilibrium (HWE) in the control group; however, in the ccRCC group, we observed deviation from HWE for rs4794976 (*LGALS9*), with an overrepresentation of GG homozygotes (f = 0.13, *p* = 0.046). For other SNPs in the ccRCC group, there was no deviation from HWE.

The overall analysis of genotype and allele distribution for all of the studied SNPs in ccRCC patients and controls is presented in Table 1. We found that the genotype distribution of *LGALS9* rs4794976 differed significantly between ccRCC patients and controls (*p* = 0.049), where the GG genotype individuals had a higher risk of disease, by about two-fold (OR = 1.91; 95% CI 1.13–3.22), compared to TT individuals. In the recessive model, carriers of a T allele (TT + GT genotypes) had about a two-fold decreased risk of ccRCC as compared to GG individuals (OR = 0.54; 95% CI 0.33–0.89; *p* = 0.015). Additionally, in an allelic analysis, the rs474976 G allele was associated with increased susceptibility to ccRCC (OR = 1.28; 95% CI 1.01–1.64; *p* = 0.044). We also noticed that the presence of a G allele in rs3751093 tends to have a protective role, decreasing the susceptibility to ccRCC by about 1.8-fold (OR = 0.55; 95% CI 0.30–1.02; *p* = 0.059), while the AA genotype increased this risk.

Furthermore, we observed that rs10057302 (*TIM-3*) genotypes tend to be differently distributed between ccRCC patients and controls (*p* = 0.071, Table 1), where carriers of the rs10057302 A allele (AC + AA genotypes) had two times lower ccRCC risk (OR = 0.45; 95% CI 0.22–0.95; *p* = 0.027) than individuals with the CC genotype. In addition, an analysis of allele distribution also confirmed that allele A in rs10057302 significantly decreased the risk of ccRCC (OR = 0.44; 95% CI 0.21–0.90; *p* = 0.018). For the other studied SNPs (rs1036199, rs361497, and rs4239242), we did not observe any association with susceptibility to ccRCC in the overall analysis.

### 2.2. Association of TIM-3 and LGALS9 Polymorphisms with Clinical Features of ccRCC Patients 

After stratification by gender, we found that, similarly to the relation observed in the whole ccRCC group, the genotype distribution of rs4794976 differed between patients and controls (Table 2), and this difference was close to significance (*p* = 0.06) in female patients. Moreover, similar to observations in the whole group of patients, in the recessive model presence of the T allele (TT + GT genotypes) decreased ccRCC risk in females by about two-fold (OR = 0.49; 95% CI 0.24–0.99; *p* = 0.047), while the GG genotype increased susceptibility to ccRCC by about two-fold. An analysis of allele distribution also confirmed that the G allele in rs4794976 significantly increased the risk of ccRCC in females (OR = 1.56; 95% CI 1.14–2.52; *p* = 0.01). In male patients, we did not observe any significant differences in genotype and allele distributions between patients and controls (Appendix A). 

When we considered the age of onset in relation to the median age of onset (age of 63 years), we noticed that, in patients older than 63, the presence of the rs4794976 GG genotype increased the risk of ccRCC development by two-fold (OR = 1.97; 95% CI 1.08–3.61; *p* = 0.030). Moreover, carriers of the rs3751093 AA genotype had a higher risk of disease in that age group by 2.47 times (OR = 2.47; 95% CI 1.22–4.98; *p* = 0.011) (Appendix A).

When taking into consideration the metastatic status of ccRCC patients, we did not notice any significant differences in the genotype distribution for all of the investigated SNPs. Similarly, we did not observe any differences when stratifying by the presence of necrosis (data not presented). However, we noticed that the A allele in rs10057302 (AC + AA genotypes) was possessed more frequently in patients with tumors bigger than 70 mm compared to patients with smaller tumors (9.2% vs. 2.1%, OR = 4.39; *p* = 0.019) (Appendix A). 

### 2.3. Haplotype Analysis

SHEsis online software [31] was used to perform a haplotype analysis. We performed the haplotype analysis for *TIM-3* and *LGALS9* separately, where haplotypes with frequencies below 1% were not considered. Both of the analyses are shown, respectively, in Table 3 and Table 4. For *TIM-3* we observed three haplotypes each for both the patients and the controls. In the case of *LGALS9,* the haplotype analysis showed the presence of six haplotypes in ccRCC and eight haplotypes in controls. 

The global distribution of the *TIM-3* haplotypes differed significantly between ccRCC patients and controls (global χ2 = 6.18, df = 2, *p* = 0.046) (Table 3), but after applying the Bonferroni correction this association lost significance. We noticed that the A A (rs1036199, rs10057302) haplotype is less frequent in patients than in controls (1.9% vs. 4.4%; *p* = 0.018). Considering this alongside the results of the genotype analysis, this points toward a protective function of the rs10057302 A allele against ccRCC development. In the case of *LGALS9*, the global distribution of haplotypes differed significantly between patients and controls (global χ2 = 18.30, df = 7, *p* = 0.011) (Table 4); however, after applying the Bonferroni correction this association lost significance. The A A C G (rs3751093, rs361497, rs4239242, and rs4794976) haplotype was more frequent in ccRCC patients than in controls (25.8% vs. 20.7%) and increased ccRCC risk (OR = 1.36; 95% CI 1.02–1.75; *p* = 0.034). Moreover, two other *LGALS9* haplotypes, A A C T and A A C T, were very rare or not detected in ccRCC compared to controls, where their frequency was above 1%. 

Furthermore, after stratification by gender, the haplotype analysis of the *TIM-3* haplotypes did not show significant differences between female patients and female controls (global χ2 = 4.72, df = 2, *p* = 0.095) (Appendix A); however, we noticed that, in female patients, the A C haplotype tends to be more frequent than in female controls (82% vs. 74%; *p* = 0.058). Moreover, the distribution of *LGALS9* haplotypes differed between female patients and female controls (global χ2 = 17.08, df = 7, *p* = 0.029), but after applying the Bonferroni correction this association lost significance (Appendix A). The A A C G haplotype was more frequent in female patients (29.4% vs. 19.6%) and increased ccRCC risk (OR = 1.68; 95% CI 1.08–2.6; *p* = 0.02), whereas the G G T T haplotype was less frequent in female patients (49.1% vs. 58.2%) and decreased ccRCC risk (OR = 0.67; 95% CI 0.46–0.98; *p* = 0.04). In the case of male patients, we did not see any significant differences in haplotype distributions (data not shown).

### 2.4. TIM-3 and LGALS9 Gene Polymorphisms in Relation to Overall Survival

*TIM-3* and *LGALS9* gene polymorphisms, as well as gender, age, stage of disease, size of tumor, and the absence or presence of metastasis as well as necrosis were subjected to an OS analysis. The analysis performed on the whole group of ccRCC patients confirmed that well-known risk factors, such as gender, stage of disease, age at diagnosis, tumor size, and the presence of metastasis as well as necrosis significantly influenced OS in our group of patients.

The results of the OS analysis are shown in Appendix A. In detail, in our ccRCC group male patients had a significantly shorter OS compared to female patients (mean ± SD: 98.11 ± 9.49 vs. 146.29 ± 17.66 months; *p* = 0.036), where females lived, on average, 4 years longer than males. As expected, the presence of metastasis and necrosis had a negative impact on OS. Patients with metastasis in other organs at the time of diagnosis lived over 8 years less than those without metastasis (mean ± SD: 42.9 ± 6.16 vs. 140.27 ± 11.89 months; *p* < 0.001). Similarly, the presence of necrosis decreased OS, and patients with no necrosis lived over 7 years longer than patients with present necrosis (mean ± SD: 95.7 ± 13.42 vs. 183.01 ± 9.99 months; *p* < 0.001). Predictably, patients above 63 years of age had a shorter OS time than their younger counterparts (mean ± SD: 109.16 ± 13.22 vs. 127.99 ± 13.95 months; *p* = 0.03). The clinical stage of disease was strongly related to OS, where mean OS for stages I, II, III, and IV was 138.4, 93.72, 91.22, and 70.57 months, respectively (*p* < 0.001). Additionally, patients with a tumor size above 70 mm lived on average 3.7 years less compared to patients with smaller tumors (mean ± SD: 62.09 ± 6.29 vs. 106.79 ± 5.67 months; *p* < 0.001). 

In addition to well-known OS-influencing factors, we also performed a correlation analysis between the studied *TIM-3* as well as *LGALS9* polymorphisms and patient OS. This analysis showed that one of the studied SNPs significantly modified OS, while the remaining ones had no association with patient OS. A detailed OS analysis is presented in Figure 2 and Appendix A. The presence of the C allele (CC + AC genotypes) in the rs1036199 of the *TIM-3* gene shortened patient OS compared to AA individuals for more than 13 months (mean ± SD: 109.21 ± 14.38 vs. 122.74 ± 11.80 months; *p* = 0.017) (Figure 2A). For rs10057302 (*TIM-3*), rs3751093 (*LGALS9*), rs361497 (*LGALS9*), rs4239242 (*LGALS9*), and rs4794976 (*LGALS9*) we did not notice any significant correlation in relation to patient OS.

### 2.5. Univariate and Multivariate Analysis of Risk Factors Influencing ccRCC Risk and Overall Patient Survival

In our study, we also performed univariate and multivariate Cox regression analyses of risk factors that influence ccRCC patient mortality. As expected, the results of the multivariate analysis confirmed the results of the univariate analysis and showed that, in addition to well-known factors associated with poorer prognoses and shorter OS (such as age at diagnosis, advanced stage of disease, the presence of metastasis and necrosis, a tumor size above 7 cm, and male gender), the presence of the C allele in the rs1036199 of *TIM-3* is associated with shorter OS (Table 5).

Next, we conducted a univariate and multivariate logistic regression analysis of the risk factors influencing ccRCC patients (versus the control group), which included all of the investigated polymorphisms of the *TIM-3* and *LGALS9* genes. Similarly, this analysis confirmed our results, showing that the possession of the rs10057302 A allele (AC + AA genotypes) decreased the risk of the development of ccRCC by more than two-fold (OR = 0.45; 95% CI 0.21–0.96; *p* = 0.039) (Appendix A).

## 3. Discussion

ccRCC stands behind one of the highest mortalities among urological cancers due to its asymptomatic nature in the early stages of the disease and a high rate of metastasis in patients at the time of diagnosis. Therefore, there is an urgent need to identify biomarkers that would identify groups at risk of this disease. The lack of proven early diagnostic markers of ccRCC emphasizes the necessity for the identification of new diagnostic and prognostic markers for ccRCC.

The etiology of ccRCC is multifactorial, but it is evident that immunosurveillance is also an important factor that influences disease risk. The proper immune response is directly related to T cell activation, which is regulated by the balance between stimulatory and inhibitory signals provided by co-signaling molecules [32]. The importance of ICs in immunosurveillance was proven by the extraordinary results of immunotherapy based on an IC blockade. Nowadays, the blockade of CTLA-4 and PD-1 or PD-L1 is an approved treatment method in many cancers, including RCC [33]. The discovery of new immunotherapy targets would improve cancer management by expanding immunotherapy effectiveness among non-responders. The panel of immune checkpoints molecules is wide and includes TIM-3, which is a regulator of both innate and adaptive immune responses. TIM-3 is expressed on IFNγ-producing CD4+ (Th1) and CD8+ (Tc1) T cells, regulatory T cells, Th17 cells, NK cells, and on innate immune cells (macrophages and dendritic cells) [34]. The dysregulation of TIM-3 has been implicated in both autoimmune diseases and cancer [9,10,11,35,36]. In several solid tumors, increased TIM-3 expression has been shown to be associated with advanced disease and poorer prognoses [37]. Importantly, TIM-3 expression is primarily observed intratumorally, with minimal expression noted in peripheral T cells and regulatory T cells (Tregs). CD8^+^PD1^+^ T cells expressing high levels of TIM-3 exhibit the most severe exhausted phenotype among tumor-infiltrating lymphocytes (TILs). CD8^+^PD1^+^TIM3^+^ T cells fail to proliferate in response to antigens and produce reduced amounts of IL-2, TNF, and IFNγ [38,39]. All of these studies highlight the importance of TIM-3 in cancer pathogenesis.

There are currently several dozen registered clinical trials investigating anti-TIM-3 antibodies as a monotherapy or in combination with other drugs, mainly anti-PD-1 antibodies, but also with chemotherapy in various solid and hematologic tumors in different clinical settings (clinicaltrials.gov). Most of them are phase I or II, and only a few studies have results (published or not published yet). It is too early to draw conclusions regarding the clinical activities and safety profiles of different anti-TIM-3 antibodies and combination strategies. This treatment strategy has to be thoroughly evaluated as it might be a clinically significant possibility for overcoming PD-1 resistance in different tumors. There is also an important question regarding the safety profiles of combined treatments, as seen with the combination of anti-PD-1 and anti-CTLA-4 blockades in the past.

Many studies have shown that polymorphisms of the *TIM-3* gene can be associated with cancer susceptibility and patient survival. Moreover, it has been shown that specific *TIM-3* SNPs can modify TIM-3 expression, influencing disease risk [24,25,26,27,28]. In a few previous studies the presence of the rs1036199 SNP in the *TIM-3* gene has been correlated with cancer risk and disease outcome. Bai et al. found that the distribution of rs1036199 genotypes differed between cases and controls. Moreover, patients carrying the rs1036199 AC genotype had a 2.81-fold higher risk of NSCLC and shorter OS than carriers of an AA genotype [24]. Similarly, Tong et al. observed that rs1036199 AC genotype and C allele carriers had increased susceptibility to pancreatic cancer. The presence of rs1036199 was also more frequent in patients with vascular infiltration than in those without [27]. In another study, Cheng et al. showed that the prevalence of the rs1036199 AC genotype and the C allele was increased in breast cancer patients compared to controls, especially in patients with metastasis, where the AC genotype was more common than in those without metastasis [26]. On the other hand, Wang et al. did not find any association of rs1036199 with breast cancer in Chinese women [28]. Wu et al. also showed that the *TIM-3* polymorphism rs1036199 may not be associated with the risk of developing epithelial ovarian cancer (EOC), nor affected its clinical outcomes [40]. In the case of RCC, the presence of the rs1036199 SNP has been shown to have a significant correlation with RCC risk in the Chinese population. Moreover, the prevalence of the rs1036199 C allele was higher in RCC patients with metastasis than in those without metastasis. The haplotype analysis showed that a haplotype containing the rs1036199 C allele, T T C (rs10053538, rs10515746, and rs1036199), was correlated with RCC risk [25].

In our study, we did not observe any correlation between rs1036199 and ccRCC risk. Nevertheless, we found the association of rs1036199 with patient OS, suggesting the involvement of rs1036199 in disease progression in our group of ccRCC patients. We found that the presence of the rs1036199 C allele significantly decreased patient OS by more than 13 months. Moreover, this result was confirmed by a Cox regression analysis. This observation is in line with observations made by Bai et al. [24]. rs1036199 is located in exon 3, which is known to encode, along with exon 4, the TIM-3 mucin domain. The presence of rs1036199 leads to the exchange of allele A with C, resulting in a missense mutation that causes the substitution of arginine (R) with leucine (L) in position 140 (R140L). Arginine is a basic amino acid carrying a positive charge, whereas leucine is a non-polar hydrophobic amino acid. Therefore, this R140L modification may have an influence on the mucin domain structure of the TIM-3 protein, which may affect the ligand’s affinity with the TIM-3 receptor, thereby altering TIM-3 downstream signaling within the cell. Moreover, it has been shown that the mucin domain contains potential sites for O-linked glycosylation [41], an important modification in protein stability and activity. The potential influence of the rs1036199 SNP on mucin domain structure may also alter the proper O-linked glycosylation of the TIM-3 protein. In spite of this, until now there have been no studies performed on the functional consequences of rs1036199 alteration. This being the case, we can only hypothesize about its true biological consequence.

Our second studied polymorphism situated in the *TIM-3* gene was rs10057302. We find that rs10057302 was significantly associated with disease risk, and that possessing the A allele (AC + AA genotypes) decreased the risk of ccRCC development by more than two times, while, conversely, the CC genotype increased the risk by two times. The rs10057302 SNP is located in intron 6 of the *TIM-3* gene. To our knowledge, rs10057302 has not been studied previously in the context of cancer or any other disease. What is interesting is that, in the subgroup analysis, we noticed that patients with a tumor size above 7 cm possessed the rs10057302 A allele more frequently than those with smaller tumors. Moreover, a haplotype analysis of *TIM-3* SNPs showed that the frequency of haplotype A A (rs1036199, rs10057302) was significantly lower in ccRCC patients than in controls. The results of a multivariate regression analysis confirmed that, among all of the studied SNPs, the rs10057302 A allele is significantly associated with decreased disease risk. Altogether, our results suggest that possessing the rs10057302 A allele may have a protective role against ccRCC development. A Kaplan–Meier analysis did not reveal any significant correlation between the presence of rs10057302 and patient survival time; however, in our cohort there is a limited number of individuals possessing the rs10057302 A allele. This being the case, studies with other groups on rs10057302‘s role in cancer are needed to confirm our observations.

In the present study, we also explored the relationships between SNPs in the gene encoding galectin-9 (*LGALS9*) and ccRCC susceptibility as well as disease outcome. Galectin-9 is a member of the galectin family of carbohydrate-binding proteins, which is characterized by the presence of two conserved carbohydrate recognition domains (CRDs) that bind galactose [42]. In humans, galectin-9 is widely distributed throughout various organ systems and tissues, with the highest expression in the spleen, stomach, colon, and lymph nodes [43]. Galectin-9 is an important pleiotropic immune modulator affecting numerous immune cell types; among others aspects, it is involved in the activation of innate immune responses [44] and the downregulation of Th17 [13] as well as Th1 responses [45]. Multiple studies have shown the capability of galectin-9 to bind to several receptors, while the best characterized is TIM-3 [18,46]. TIM-3-binding galectin-9 attenuates T cell expansion and effector functions in the tumor microenvironment [18]. Multiple studies have shown a multi-faceted role for galectin-9 that contributes to tumorigenesis via tumor cell transformation, cell cycle regulation, angiogenesis, and cell adhesion [47,48,49]. Galectin-9 expression is frequently altered in cancer and involved in several aspects of tumor progression [15,16,17,18], making galactin-9 an interesting potential prognostic marker and a therapeutic target for several malignancies.

So far, there have been no reports regarding associations between cancer and *LGALS9* polymorphisms, despite galectin-9 being documented as playing an important role in cancer pathology. The first study on *LGALS9* polymorphisms’ potential role in disease came from Rosen et al., who examined the association of *LGALS9* gene variants with the development of advanced alcoholic liver disease (ALD). In this study, Rosen’s group found an association between four *LGALS9* SNPs (rs732222, rs3751093, rs4239242, and rs4794976) and the risk of ALD [50]. In our study, we documented that the genotype distribution of rs4794976 differs significantly between ccRCC patients and controls, where the presence of the GG genotype increased the risk of disease by about 1.9-fold compared to the AA genotype. Moreover, we observed that the GG genotype significantly increased the risk of ccRCC in women (but not in men) and in patients older than 63 years old. Similar to our results, in ALD rs4794976 genotype frequency differed between studied groups, with an over-representation of rs4794976 G allele carriers (GG + GT genotypes) among individuals that developed ALD [50]. On the contrary, Xu’s group, investigating the association of *LGALS9* polymorphisms with rheumatoid arthritis (RA), showed that rs4794976 allele T as well as TT and TT + TG genotypes were significantly associated with RA risk [30]. This difference may occur due to the different pathologies: RA is an autoimmune and inflammatory disease, whereas in cancer the immune system response is suppressed. This, in turn, would explain the opposite results in regard to rs4794976 occurrence. As we mentioned in our results for this SNP, we observed deviation from HWE in the patient group, while the control group was in complete HWE. This fact may confirm the association between rs4794976 and ccRCC risk, since, according to Lee et al., in the presence of an association with disease, cases do not need to be in HWE, and deviation from HWE of datasets of affected individuals is sufficient to discover relationships with disease [51].

Furthermore, we observed that the presence of the rs3751093 G allele may have a protective role, decreasing susceptibility to ccRCC by 1.8-fold. This observation was also seen in a subgroup analysis when age of onset was considered. Patients older than 63 years of age possessing the G allele were less susceptible to disease than patients with AA homozygotes. In RA, Xu et al. documented the decreased frequency of the rs3751093 GA genotype in RA patients compared to controls, suggesting its protective role in RA [30]. Additionally, Rosen’s group noticed that the rs3751093 GG genotype was less frequent in individuals prone to developing ALD compared to subjects who were protected from developing ALD. Moreover, the expression levels of galectin-9 transcripts were lower in PBMCs treated with ethanol carrying the rs3751093 GG genotype compared to PBMCs treated with ethanol carrying rs3751093 AA and AG genotypes [50]. Further studies on rs3751093′s role in cancer are needed to confirm our results.

For rs4239242, in RA there are two studies reporting distinct results. In Vilar’s study, the rs4239242 TT genotype was positively correlated with the incidence of RA, and the TC genotype was more frequent in controls than in RA patients [29], whereas Xu’s group did not report any significant relationship between rs4239242 and RA [30]. Additionally, in ALD the rs4239242TT genotype was associated with a lower risk of developing ALD. Moreover, PBMCs carrying the rs4239242 TT genotype showed lower levels of galectin-9 transcripts after ethanol stimulation compared to CC and CT genotypes [50]. In our study, we did not observe any correlation of rs4239242 with ccRCC risk. Differences in reported results probably arise due to different types of studied diseases, where specific polymorphisms can have diverse effects on disease pathogenesis.

The association between *LGALS9* SNPs and patient survival has not been studied previously by others. Our Kaplan–Meier analysis on the influence of investigated *LGALS9* SNPs on patient OS in ccRCC did not reveal any significant correlation. Thus, in our study, *LGALS9* SNPs did not show any association with ccRCC progression.

Our haplotype analysis showed that the A A C G (rs3751093, rs361497, rs4239242, and rs4794976) haplotype was more common in patients and may be considered as a risk factor for ccRCC, whereas A A C T appeared to have a protective role. In RA, the G T G C G (rs3751093, rs4239242, rs4794976, rs4795835, and rs732222) haplotype was less prevalent in RA patients compared to controls, while the G T T C G haplotype was positively correlated with RA risk [30]. Alternatively, in ALD, the G T G G T haplotype (rs3751093, rs4794976, rs4239242, rs3763959, and rs732222) was negatively correlated with ALD, while G C G G T was positively correlated with ALD risk [50]. Additionally, in this case, differences in haplotype frequencies between studies would arise from different types of studied diseases, which make it impossible to directly compare results.

In our present work, we also conducted univariate and multivariate logistic regression analyses of risk factors influencing ccRCC patients, including all of the investigated polymorphisms. These analyses showed that possessing the rs10057302 A allele (AC + AA genotype) decreases the risk of the development of ccRCC. Furthermore, we performed univariate and multivariate Cox regression analyses of risk factors that influence the mortality of ccRCC patients. As expected, several clinical factors, such as female gender, young age, no malignancy, lack of necrosis, early disease stage, and a tumor size below 7 cm, were associated with better OS of ccRCC patients in both the univariate and multivariate analyses. Moreover, these analyses confirmed our finding that the presence of the rs1036199 AA genotype is a risk factor influencing patient OS.

Limitations of our study are the lack of clinical data for the control group as well as mismatched ages of patients and controls. For the control group, we only have data on gender, age, and lack of cancer diseases. For some controls we also have data about smoking and some anthropometrical data which were not relevant to the study. We realize that the control group is not matched in relation to age, and that healthy individuals could develop cancer in future; however, the incidence of renal cell cancer is, on average, about 1 per 10,000 cases (different in men and women). In light of this, the chance that in the control group there would be a significant number of people who at a later age would develop cancer and distort the obtained results is small. Another limitation was the inability to investigate the functional role of studied polymorphisms and their effect on protein expression. On the other hand, the strengths of this study include the long period of patient observation, which lasted more than 10 years. Additional research into the underlying mechanisms influenced by specific SNPs has to be elicited to further confirm our findings. Finally, studies on larger groups of patients as well as on other populations are needed to confirm our findings.

## 4. Conclusions

For the first time, we showed that SNPs of the gene encoding galectin-9 could be associated with susceptibility to cancer. In particular, rs4794976 of the *LGALS9* gene may be considered a low penetrating risk factor for the development of ccRCC. Additionally, we found that rs10057302 of *TIM-3* can have a protective role in ccRCC, whereas rs1036199 of the *TIM-3* gene showed a negative correlation with ccRCC progression. Moreover, there was evidence suggesting that variants of rs4794976, as well as rs10057302, may also relate to the risk of ccRCC in females and older patients. In conclusion, our study showed an association of *TIM-3* and *LGALS9* polymorphisms with ccRCC risk and outcomes; however, extended studies on larger groups of patients and the functional evaluation of studied SNPs are needed to confirm our results.

## 5. Materials and Methods

### 5.1. ccRCC Patients

The group of patients enrolled in this study consisted of 237 ccRCC patients (151 male and 86 females) diagnosed at the Department of Urology and Oncologic Urology at Wroclaw Medical University. Patients were diagnosed between 2009 and 2012, while samples were collected within a period of 2010 to 2012. The studies involving human participants were reviewed and approved by the Bioethical Committee of Wroclaw Medical University. The DNA used in the presented study was isolated from patients recruited for the previous project approved by the Ethics Committee of Wroclaw Medical University (KB 55/2010). For the purpose of this study (reuse of the material), we obtained additional approvals from the Ethics Committee of Wroclaw Medical University (KB 587/2020 and KB 755/2022). Patients provided their written informed consent to participate in this study. Overall survival was assessed from the date of surgery to the date of death from any cause or up to 24 January 2020, when data collection was completed. Patients’ characteristics are presented in Table 6.

### 5.2. Controls

The control group comprised 410 (258 males and 148 females) subjects from the same geographic region as ccRCC patients. Blood samples from healthy subjects were collected by the Wrocław Blood Bank or donated by employees of the Ludwik Hirszfeld Institute of Immunology and Experimental Therapy. The studies involving human participants were reviewed and approved by the Bioethical Committee of Wrocław Medical University, Wrocław, Poland. Participants provided their written informed consent to participate in this study.

### 5.3. SNP Selection

Four of six selected SNPs, rs1036199 (*TIM-3*), rs3751093 (*LGALS9*), rs4239242 (*LGALS9*), and rs4794976 (*LGALS9*), have been studied previously by other groups in the context of cancer risk (only *TIM-3* SNPs) and RA risk (*TIM-3* and *LGALS9* SNPs). In this study we also selected two new previously unstudied SNPs, rs10057302 (*TIM-3*) and rs361497 (*LGALS9*), using the UCSC database, available at https://genome.ucsc.edu/index.html accessed on 27 November 2022. The localization of each SNP is shown in Figure 1, and extended information about each SNP is provided in Appendix A.

### 5.4. DNA Isolation and SNP Genotyping

Genomic DNA was isolated from refrozen blood samples by an Invisorb Spin Blood Mini Kit (Stratec Molecular, Berlin, Germany) or a QIAamp DNA Blood Mini Kit (Qiagen, Hilden, Germany) according to the manufacturers’ instructions. Three SNPs were genotyped using TaqMan Genotyping Master Mix (Applied Biosystems, Frederic, MD, USA) and TaqMan assays. ID: rs1036199 (*TIM-3*) C___2082038_1_, rs10057302 (*TIM-3*) C__29607693_10, and rs4794976 (*LGALS9*) C__29024730_10. All reactions were run on a ViiA7 Real-Time PCR System (Applied Biosystems, Singapore, Singapore).

SNP rs3751093 (*LGALS9*) was genotyped using the tetra-primer amplification refractory mutation system–polymerase chain reaction (ARMS–PCR). Primers were designed using Primer1 online software (http://primer1.soton.ac.uk/primer1.html, accessed on 11 February 2022). The primers used in ARMS–PCR include forward inner primer: 5′-GCGGCGGAGAGATGGCCTTCATCA-3′; reverse inner primer: 5′-ACTCAGGTAGGGAGCCTGGGATCC-3′; forward outer primer: 5′- GCTGGGAGTGCCTACTTCCCTCTGTG-3′; and reverse outer primer: 5′- GTTCTCTTTGGGATGCCCCCACCC-3′. The PCR was prepared in a volume of 10 µL with 100–150 ng of DNA, a mix of primers in a 1:1:5:5 ratio (F_out:R_out:F_in:R_in), 0.2 Mm Dntp Mix (Thermo Scientific, Vilnus, Lithuania), and DreamTaq Green DNA Polymerase (Thermo Scientific, Vilnus, Lithuania). All of the reactions were run on a T100 Thermal Cycler (BioRad, Singapore, Singapore). The protocol used in the thermal cycler was as follows: initial denaturation at 95 °C for 3 min, followed by 32 cycles of denaturation at 95 °C for 30 s per cycle, combined annealing and extension for 1 min at 72 °C, and final extension for 5 min at 72 °C. Products of PCR reactions were separated on 2% agarose gel and visualized with the help of a UV trans-illuminator VOO 7237 (Vilber Lourmat, Marne la Valee, France.

SNPs rs361497 (*LGALS9*) and rs4239242 (*LGALS9*) were genotyped using polymerase chain reaction-restriction fragment length polymorphism (PCR-RFLP). Rs361497 was detected using 5′-TGCCTGCCTGGTCTCTC-3′ (forward primer) and 5′-GGTCACTGTGGCAGTGGT-3′ (reverse primer), and digested using BglI restriction enzyme (Thermo Scientific, Vilnus, Lithuania). Rs4239242 was detected using 5′-CGATGCCTTTCATCACCACCA-3′ (forward primer) and 5′-CACCTCCTTCTTGGGTCTGAT-3′ (reverse primer), and digested using EcoRI Fast Digest restriction enzyme (Thermo Scientific, Vilnus, Lithuania). The PCR was prepared in a volume of 10 µL with 50–100 ng of DNA, 400 Nm concentration of each primer (forward and reverse), 0.2 Mm Dntp Mix (Thermo Scientific, Vilnus, Lithuania), and Taq DNA Polymerase (EurX, Gdansk, Poland). All reactions were run on At100 Thermal Cycler (BioRad, Singapore, Singapore). The protocol used in the thermal cycler was as follows: initial denaturation at 95 °C for 3 min, followed by 35 cycles of denaturation at 95 °C for 30 s per cycle, annealing at 60 °C for 45 s and extension for 1 min at 72 °C, and final extension for 5 min at 72 °C. PCR products were next digested with selected restriction enzymes. RFLP digestion was carried out in a volume of 15 Μl with 1.5 units of the restriction enzyme and 5–10 Μl (0.1–0.5 μg) of PCR product in O buffer (BglI) or Fast Digest buffer (EcoRI FD) (Thermo Scientific, Vilnus, Lithuania) for 1.5 h (BglI) or 0.5 h (EcoRI FD) at 37 °C. Digested products were separated on 2% agarose gel and visualized with the help of a UV trans-illuminator VOO 7237 (Vilber Lourmat, Marne la Valee, France).

### 5.5. Statistical Analyses

Statistical analyses were performed using Statistica 13.1 (TIBCO, Inc., Palo Alto, CA, USA) and PQStat v.1.8.0.476 software (Poznan, Poland). For measurable variables, the means, medians, and standard deviations were calculated. All of the investigated quantitative variables were checked with the Shapiro–Wilk test. For all of the genotyped *TIM-3* and *LGALS9* polymorphisms the evaluation of Hardy–Weinberg equilibrium (HWE) was performed independently for ccRCC patients and healthy controls by comparing the observed and expected frequencies of genotypes by using the χ2 test. The χ2 test was used to compare categorical data between ccRCC patients and controls. Odds ratios (ORs) and 95% confidence intervals (95% CIs) were calculated using a binary logistics regression model to evaluate the relationship between studied polymorphisms and susceptibility to ccRCC. Haplotype frequencies for pairs of alleles were determined using the online software SHEsis [31,52], where haplotypes with frequencies below 0.01 were not considered. Differences between groups were considered statistically significant if *p* < 0.05.

Survival analysis (OS) was performed using a Kaplan–Meier estimator in SigmaPlot 11.0 software (Systat Software, San Jose, CA, USA). The log-rank test was used to compare patient survival against selected clinical variables.

Univariate and multivariate Cox proportional hazard regression models were used to investigate factors associated with the mortality of ccRCC patients. Independent variables examined included age at diagnosis, stage of disease (II, III, and IV, ref. I), metastasis (present, ref. no), necrosis (present, ref. no), tumor size (>70 mm, ref. ≤ 70 mm), sex (ref. female), rs1036199 (AC + CC, ref. AA), rs10057302 (AC + AA, ref. CC), rs3751093 (AG + AA, ref. GG), rs361497 (AG + AA, ref. GG), rs4239242 (CT + CC, ref. TT), and rs4794976 (CT + GG, ref. TT). Variables with *p* < 0.20 in the univariate analysis were entered into the multivariate model. Variables that were found to be significant (*p* < 0.05) in both the univariate and multivariate analyses were considered to be factors associated with mortality.

## Figures and Tables

**Figure 1 ijms-24-02042-f001:**
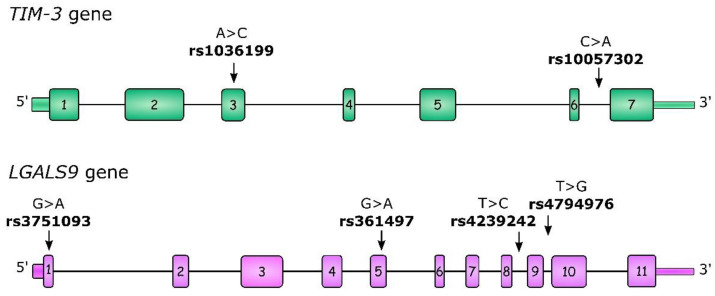
Structure of the *TIM-3* and *LGALS9* genes as well as the localization of studied single-nucleotide polymorphisms (SNPs). Boxes indicate exons and lines indicate introns, 5′UTR and 3′UTR regions.

**Figure 2 ijms-24-02042-f002:**
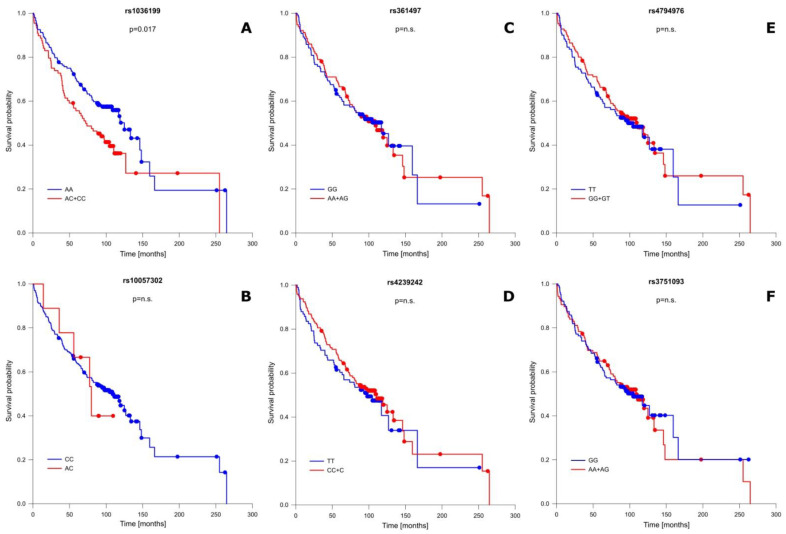
Probability of survival in relation to *TIM-3* and *LGALS9* gene polymorphisms: (**A**) rs1036199 (*TIM-3*); (**B**) rs10057302 (*TIM-3*); (**C**) rs361497 (*LGALS9*); (**D**) rs4239242 (*LGALS9*); (**E**) rs4794976 (*LGALS9*); and (**F**) rs3751093 (*LGALS9*). n.s.—not significant.

**Table 1 ijms-24-02042-t001:** Genotype and allele distribution of *TIM-3* and *LGALS9* SNPs among ccRCC patients and controls.

			Cases		Controls					
SNP	Genotype	Allele	N	%	N	%	OR	95% CI		*p*-Value
**rs1036199**										
	AA		149	62.87	256	62.44	1			0.377
	AC		83	35.02	137	33.41	1.042	0.743	1.462	
	CC		5	2.11	17	4.15	0.539	0.203	1.435	
	AC + CC		88	37.13	154	37.56	0.983	0.707	1.367	0.913
	AA + AC		232	97.89	393	95.85	1.880	0.711	4.968	0.169
		A	381	80.38	649	79.15	1			
		C	93	19.62	171	20.85	0.928	0.700	1.230	0.596
**rs10057302**										
	CC		228	96.20	376	91.71	1			0.071
	AC		9	3.80	32	7.80	0.482	0.229	1.011	
	AA		0	0.00	2	0.49	0.330	0.016	6.895	
	AC + AA		9	3.80	34	8.29	0.454	0.217	0.948	**0.027**
	CC + AC		237	100.00	408	99.51	2.907	0.139	60.809	0.282
		C	465	98.10	784	95.61	2			
		A	9	1.90	36	4.39	0.439	0.213	0.904	**0.018**
**rs3751093**										
	GG		128	54.47	242	59.02	1			0.142
	AG		86	36.60	147	35.85	1.107	0.787	1.556	
	AA		21	8.94	21	5.12	1.887	1.000	3.561	
	AG + AA		107	45.53	168	40.98	1.204	0.872	1.663	0.261
	GG + AG		214	91.06	389	94.88	0.551	0.296	1.024	**0.059**
		G	342	72.77	631	76.95	1			
		A	128	27.23	189	23.05	1.250	0.964	1.621	0.093
**rs361497**										
	GG		120	51.06	225	55.42	1			0.220
	AG		93	39.57	157	38.67	1.111	0.792	1.558	
	AA		22	9.36	24	5.91	1.719	0.930	3.174	
	AG + AA		115	48.94	181	44.58	1.191	0.863	1.642	0.287
	GG + AG		213	90.64	382	94.09	0.608	0.335	1.103	0.103
		G	333	70.85	607	74.75	1			
		A	137	29.15	205	25.25	1.219	0.946	1.571	0.128
**rs4239242**										
	TT		91	38.56	177	43.17	1			0.307
	CT		104	44.07	178	43.41	1.136	0.801	1.610	
	CC		41	17.37	55	13.41	1.451	0.902	2.332	
	CT + CC		145	61.44	233	56.83	1.209	0.872	1.675	0.252
	TT + CT		195	82.63	355	86.59	0.735	0.474	1.140	0.174
		T	286	60.59	532	64.88	1			
		C	186	39.41	288	35.12	1.202	0.951	1.518	0.124
**rs4794976**										
	TT		110	46.61	210	51.22	1			**0.049**
	GT		92	38.98	166	40.49	1.058	0.751	1.491	
	GG		34	14.41	34	8.29	1.905	1.127	3.221	
	GT + GG		126	53.39	200	48.78	1.202	0.873	1.656	0.260
	TT + GT		202	85.59	376	91.71	0.538	0.326	0.888	**0.015**
		T	312	66.10	586	71.46	1			
		G	160	33.90	234	28.54	1.285	1.007	1.638	**0.044**

**Table 2 ijms-24-02042-t002:** Genotype and allele distribution of *TIM-3* and *LGALS9* SNPs among female patients and female controls.

			Cases		Controls					
SNP	Genotype	Allele	N	%	N	%	OR	95% CI		*p*-Value
**rs1036199**										
	AA		58	67.44	91	61.49	1			0.191
	AC		28	32.56	52	35.14	0.849	0.484	1.489	
	CC		0	0.00	5	3.38	0.142	0.008	2.620	
	AC + CC		28	32.56	57	38.51	0.775	0.445	1.352	0.362
	AA + AC		86	100.00	143	96.62	-	-	-	-
		A	144	83.72	234	79.05	1			
		C	28	16.28	62	20.95	0.740	0.454	1.207	0.217
**rs10057302**										
	CC		83	96.51	134	90.54	1			-
	AC		3	3.49	14	9.46	0.389	0.117	1.289	
	AA		0	0.00	0	0.00	-	-	-	
	AC + AA		3	3.49	14	9.46	0.389	0.117	1.289	0.090
	CC + AC		86	100.00	148	100.00	-	-	-	-
		C	169	98.26	282	95.27	1			
		A	3	1.74	14	4.73	0.402	0.123	1.313	0.096
**rs3751093**										
	GG		41	48.24	90	60.81	1			0.137
	AG		34	40.00	48	32.43	1.551	0.877	2.745	
	AA		10	11.76	10	6.76	2.181	0.860	5.533	
	AG + AA		44	51.76	58	39.19	1.659	0.971	2.834	0.063
	GG + AG		75	88.24	138	93.24	0.545	0.222	1.341	0.190
		G	116	68.24	228	77.03	1			
		A	54	31.76	68	22.97	1.561	1.025	2.375	**0.038**
**rs361497**										
	GG		39	45.88	85	57.43	1			0.132
	AG		35	41.18	53	35.81	1.436	0.814	2.534	
	AA		11	12.94	10	6.76	2.371	0.947	5.935	
	AG + AA		46	54.12	63	42.57	1.585	0.929	2.704	0.090
	GG + AG		74	87.06	138	93.24	0.491	0.203	1.187	0.113
		G	113	66.47	223	75.34	1			
		A	57	33.53	73	24.66	1.541	1.020	2.327	**0.040**
**rs4239242**										
	TT		30	35.29	60	40.54	1			0.309
	CT		35	41.18	65	43.92	1.075	0.592	1.953	
	CC		20	23.53	23	15.54	1.730	0.830	3.609	
	CT + CC		55	64.71	88	59.46	1.244	0.718	2.155	0.429
	TT + CT		65	76.47	125	84.46	0.598	0.308	1.162	0.131
		T	95	55.88	185	62.50	1			
		C	75	44.12	111	37.50	1.315	0.897	1.928	0.161
**rs4794976**										
	TT		36	42.35	83	56.08	1			0.060
	GT		31	36.47	48	32.43	1.486	0.821	2.690	
	GG		18	21.18	17	11.49	2.418	1.130	5.174	
	GT + GG		49	57.65	65	43.92	1.729	1.011	2.955	**0.044**
	TT + GT		67	78.82	131	88.51	0.486	0.237	0.994	**0.047**
		T	103	60.59	214	72.30	1			
		G	67	39.41	82	27.70	1.696	1.139	2.525	**0.009**

**Table 3 ijms-24-02042-t003:** Haplotype distribution of *TIM-3* SNPs between ccRCC patients and controls.

Haplotype *	ccRCC (%)	Control (Freq)	Odds Ratio [95% CI]	*p*-Value
A A	9.00 (1.9)	35.95 (4.4)	0.421 [0.201~0.882]	0.018
A C	372.00 (78.5)	611.05 (74.7)	1.235 [0.943~1.617]	0.125
C C	93.00 (19.6)	170.95 (20.9)	0.924 [0.697~1.225]	0.583
Global χ2 = 6.18, df = 2, *p* = **0.046**				

* rs1036199, rs10057302. Significant results are bolded.

**Table 4 ijms-24-02042-t004:** Haplotype distribution of *LGALS9* SNPs between ccRCC patients and controls.

Haplotype *	ccRCC (%)	Control (%)	Odds Ratio [95% CI]	*p*-Value
A A C G	120.88 (25.8)	168.59 (20.7)	1.335 [1.021~1.746]	**0.034**
A A C T	0.00 (0)	11.03 (1.4)	-	**0.011**
G A C G	11.02 (2.4)	9.88 (1.2)	1.964 [0.826~4.672]	0.120
G A T T	1.05 (0.2)	13.43 (1.6)	0.134 [0.018~0.980]	**0.020**
G G C G	11.86 (2.5)	20.90 (2.6)	0.987 [0.480~2.032]	0.972
G G C T	40.24 (8.6)	67.47 (8.3)	1.041 [0.692~1.567]	0.846
G G T G	14.24 (3.0)	28.41 (3.5)	0.868 [0.455~1.658]	0.668
G G T T	262.59 (56.1)	483.95 (59.3)	0.870 [0.690~1.098]	0.241
Global χ2 = 18.30, df = 7, *p* = **0.011**				

* rs3751093, rs361497, rs4239242, and rs4794976. Significant results are bolded.

**Table 5 ijms-24-02042-t005:** Univariate and multivariate Cox regression analysis of risk factors’ influences on the mortality of ccRCC patients.

		Univariate					Multivariate		
Variable		HR	95% CI		*p*-Value	HR	95% CI		*p*-Value
Age at diagnosis		1.03	1.01	1.05	**0.002**	1.03	1.01	1.06	**0.007**
Stage of disease (ref. I)	II	0.29	0.14	0.63	**0.001**	-	-	-	-
	III	3.68	2.17	6.25	0.502	-	-	-	-
	IV	34.31	14.38	81.88	**<0.001**	-	-	-	-
Metastasis (ref. no)	Present	4.80	3.23	7.13	**<0.001**	3.64	2.22	5.94	**<0.001**
Necrosis (ref. no)	Present	2.61	1.74	3.89	**<0.001**	1.63	1.01	2.63	**0.048**
Tumor size (ref. ≤ 70)	>70 mm	2.48	1.69	3.64	**<0.001**	1.73	1.06	2.82	**0.027**
Sex (ref. female)	Male	1.51	1.02	2.21	**0.038**	1.79	1.08	2.97	**0.023**
rs1036199 (ref. AA)	AC + CC	1.54	1.08	2.19	**0.017**	1.96	1.25	3.07	**0.003**
rs10057302 (ref. CC)	AC + AA	1.15	0.47	2.83	0.753	-	-	-	-
rs3751093 (ref. GG)	AG + AA	0.98	0.69	1.39	0.908	-	-	-	-
rs361497 (ref. GG)	AG + AA	0.89	0.62	1.27	0.511	-	-	-	-
rs4239242 (ref. TT)	CT + CC	0.91	0.64	1.30	0.608	-	-	-	-
rs4794976 (ref. TT)	GT + GG	1.02	0.72	1.45	0.918	-	-	-	-

**Table 6 ijms-24-02042-t006:** Characteristics of the ccRCC group.

Variable	All N = 237		Male N = 151		Female N = 86	
**Age at Diagnosis**						
Median	62		61		63	
Mean	62.61		62.01		63.67	
Q1–Q3	56–70		56–68		58–71	
Min., max.	21, 85		21, 85		24, 85	
**BMI**						
Median	27.7		27.7		27.75	
Mean	28.29		28.26		28.33	
Q1–Q3	24.6–31.5		25.1–30.7		23.85–31.2	
Min., max.	19.1, 43.8		19.7, 43.8		19.1, 43.8	
**Stage of Disease**	**N**	**%**	**N**	**%**	**N**	**%**
I	108	(45.57)	63	(41.72)	45	(52.33)
II	26	(10.97)	20	(13.25)	6	(6.98)
III	26	(10.97)	16	(10.60)	10	(11.63)
IV	76	(32.07)	51	(33.77)	25	(29.07)
Unknown	1	(0.42)	1	(0.66)	0	(0)
**Metastasis**						
No	165	(69.62)	101	(66.89)	64	(74.42)
Present	53	(22.36)	35	(23.18)	18	(20.93)
Unknown	19	(8.02)	15	(9.93)	4	(4.65)
**Necrosis**						
No	118	(59.00)	71	(55.47)	47	(65.28)
Present	82	(41.00)	57	(44.53)	25	(34.72)
Unknown	0	(0)	0	(0)	0	(0)
**Tumor Size**						
<70 mm	143	60.34	87	(57.61)	56	(65.12)
>70 mm	65	27.42	48	(31.79)	17	(19.77)
Unknown	29	12.24	16	(10.60)	13	(15.11)

Stage of disease according to the 2009 TNM system, grading according to Fuhrman classification.

## Data Availability

The original contributions are presented within the article and in the Appendix A. Additional data are available upon reasonable request from the corresponding author.

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
