# Peer review of "The Variations’ in Genes Encoding TIM-3 and Its Ligand, Galectin-9, Influence on ccRCC Risk and Prognosis"

_ijms, 2023, doi:10.3390/ijms24032042_

Round 1

Reviewer 1 Report

Thank you for authors for this paper. 

Some suggestions/corrections.

Some language corrections are needed, for example line 84 Chines->Chinese, Paragraphs beginning from line 88 and 306: some word order problems. Please check out throughout the paper.

Line 199 and forwards when speaking about survival. Please add units (months?)

I prefer concentrating on malignant diseases with correlations of these genes, not ALD and RA, it is just confusing to a reader. At page 11 there is a lot of unnecessary text. You might just say (if there was no correlation with cancer) that previously these genotype has been correlated with xx disease.

There are several TIM-antibodies already in clinical trials. Please correlate these results with them. It seems that TIM-antibody as a single treatment does not improve survival or progression-free survival. These have to be given with PD-(L)1-inhibitors. Do you have any explanation to this? 

The lack of clinical data for the control group is a problem, as case-control -studies are normally matched. You should discuss more the effects of this.

Please also discuss more the functional role, is this testing your next step? Is it possible to done at all, etc?

What is your main clinical point or is it possible to have one? Are these gene polymorphism test available in the clinic or should they be?

Please clarify in the lines 432 and 434: are these tumor samples and tumor DNA?

Reviewer 2 Report

The paper is interesting and provides a useful contribution to its area of research. The calculated examples are very interesting. Before publishing the author has to make improvements.

At the end of the article should be the CONCLUSION, where the results are presented.

Reviewer 3 Report

The authors represent a comprehensive statistical analysis of how gene variations encoding TIM-3 and galectin- 2 influence ccRCC risk and prognosis.

There are two major concerns:

11. Statement of “rs4794976 T allele (GT+TT genotypes) decreased susceptibility to ccRCC by two-fold compared to corresponding homozygotes”. From the sentence in row 106 is clear that there is a deviation from HWE for the rs479497.

It is well known that Hardy Weinberg Equilibrium Disturbances in Case-Control Studies lead to non-conclusive results. The authors have to check if factors such as inbreeding caused by consanguinity, population stratification, or technical problems in genotyping are present. Since the control population follows an HWE, the presence of HWD among ccRCC patients might be explained by the genetic association and evidencing a real link between the locus and the trait under study.

Based on the check, all text within the manuscript about rs4794976 (T-allele) has to be reconsidered, clarified, or changed.

22. The present study involves DNA sample isolated in the previous study. Recent new approvals from the Ethics Committee of Wroclaw Medical University (KB 587/2020) and (KB 755/2022) exist. How did the patients provide their written informed consent to participate in the recent study, since some died during 2010 - 2022? Please, explain.

Minor:

11. Row 18-19: “con-tain-ing-3”, should be “containing-3”

22. Row 465: “TaqMan assays”, should be “TaqMan assays ID”

Reviewer 4 Report

This manuscript can be accepted.

Author Response

We thank the Reviewer for such opinion.

Round 2

Reviewer 1 Report

Thank you for the corrections.

The last question was about the next text: "Patient samples were collected during the period from 2009 432 to 2012. The studies involving human participants were reviewed and approved by the 433 Bioethical Committee of Wroclaw Medical University. DNA used in the presented study 434 was isolated from patients recruited..." These should be clarified.

All the changes did not have a red colour, so it was a pretty hard to follow what was done altogether.

I still propose, that there should be some clinical points in the discussion, as written in the answer to TIM-antibody treatment in the response letter.

Also there was not discussed more in the text the meaning of lack of clinical data as was done in the response letter. This should be added.

Author Response

Response to the comments provided by REVIEWER 1.

The last question was about the next text: "Patient samples were collected during the period from 2009  to 2012. The studies involving human participants were reviewed and approved by the  Bioethical Committee of Wroclaw Medical University. DNA used in the presented study 434 was isolated from patients recruited..." These should be clarified.

Thanks the Reviewer for noticing our inconsistency.

Please allow us to present more detailed explanation which we provided also for the Reviewer 3. 

Our study on the associations of the immune check-points genes variations influence on RCC risk started in 2010 (published in: Partyka A, Tupikowski K, Kolodziej A, Zdrojowy R, Halon A, Malkiewicz B, Dembowski J, Frydecka I, Karabon L. Association of 3' nearby gene BTLA polymorphisms with the risk of renal cell carcinoma in the Polish population. Urol Oncol. 2016 Sep;34(9):419.e13-9; Tupikowski K, Partyka A, Kolodziej A, Dembowski J, Debinski P, Halon A, Zdrojowy R, Frydecka I, Karabon L.CTLA-4 and CD28 genes' polymorphisms and renal cell carcinoma susceptibility in the Polish population--a prospective study. Tissue Antigens. 2015 Nov;86(5):353-61) and we received approval from local Ethics Committee (Medical University of Wroclaw- KB- 55/2010) for that study. Then, in 2020 we asked Ethics Committee of Wroclaw Medical University for the use of anonymized archival material for next study (published in: Marta Wagner, Monika Jasek and Lidia Karabon, Immune checkpoint molecules - inherited variations as markers for cancer risk. Frontiers in Immunology, 2021, 14;11:606721. doi: 10.3389/fimmu.2020.606721) (KB 587/2020) and then in 2022 Ethics Committee broaden that approval for current investigations (KB 755/2022).

As we states in our first publications from that project that started from 2010. “ We prospectively recruited 323 kidney tumor patients treated at the Department of Urology and Oncological Urology, Wroclaw Medical University between March 2010 and November 2012. Of those patients, 13 were eliminated due to insufficient data or withdrawal of consent.”  (Tupkowski et al. CTLA-4 and CD28 genes' polymorphisms and renal cell carcinoma susceptibility in the Polish population--a prospective study. Tissue Antigens. 2015 Nov;86(5):353-61. doi: 10.1111/tan.12671).

However, some patients were diagnosed in 2009, while included in the study in 2010.

This is why the date of diagnosis appeared in the patients' descriptions instead of the date of study recruitment. In the revised manuscript, this date has been changed.

On the Reviewer 3 request we provided 11 scanned copy of  randomly selected written concerns signed in 2010-2012 years from RCC patients.

All the changes did not have a red colour, so it was a pretty hard to follow what was done altogether.

According to reviewer suggestion all chances are marked in red.  

I still propose, that there should be some clinical points in the discussion, as written in the answer to TIM-antibody treatment in the response letter.

 In the response on the Reviewer suggestion we added the following paragraph to the Discussion

“There are currently several dozen registered clinical trials investigating anti-TIM-3 antibodies as monotherapy or in combination with other drugs mainly anti-PD-1 anti-bodies but also with chemotherapy in various solid and hematologic tumors in different clinical settings (clinicaltrials.gov). Most of them are phase I or II and only few studies have results (published or not published yet). It is too early to draw conclusions re-garding clinical activity and safety profile of different anti-TIM-3 antibodies and com-bination strategies. This treatment strategy has to be thoroughly evaluated thou as it might be clinically significant possibility of overcoming PD-1 resistance in different tumors. There is also an important question regarding safety profile of combined treatment as seen with combination of anti-PD-1 and anti CTLA-4 blockade in the past”.

Also there was not discussed more in the text the meaning of lack of clinical data as was done in the response letter. This should be added.

According to Reviewer’s  suggestion we added to the part describing limitation of the study the explanation similar to that provided in the first response to the reviewer comments

It is as follow.

“The limitation of our study is the lack of clinical data for the control group, as well as mis-matched age of patients and controls. For the control group, we only have data on gender, age, and about the lack of cancer diseases. For some controls we also have data about smoking and some anthropometrical data which were not relevant in that study. We realize that control group is not matched in relation to age and healthy individuals could develop cancer in future. However, the incidence of renal cell cancer is in average about 1 per 10000 cases (different in men and women). In view of this, the chance that in the control group there would be a number of people at a later age that would distort the obtained results is small. “

Reviewer 3 Report

I agree with the authors' responses and still have an ethical concern.

The DNA samples were isolated in 2010 in connection to another project, and the authors correctly received two new ethical permissions. I would like the authors to send a scanned copy of 10 randomly selected written concerns from patients signed in 2010.

Author Response

Please , see the attachment

-------------To avoid leakage of patient information, the patient's signature is removed by the editorial office.
